# Class-Incremental Learning via Dual Augmentation

**Fei Zhu**[1,2], **Zhen Cheng**[1,2], **Xu-Yao Zhang**[1,2*], **Cheng-Lin Liu**[1,2,3]

[1]NLPR, Institute of Automation, Chinese Academy of Sciences, Beijing 100190, China
[2]University of Chinese Academy of Sciences, Beijing, 100049, China
[3]Center for Excellence of Brain Science and Intelligence Technology, CAS
{zhufei2018, chengzhen2019}@ia.ac.cn, {xyz, liucl}@nlpr.ia.ac.cn

## Abstract

Deep learning systems typically suffer from catastrophic forgetting of past knowledge when acquiring new skills continually. In this paper, we emphasize two dilemmas, representation bias and classifier bias in class-incremental learning, and present a simple and novel approach that employs explicit *class augmentation (classAug)* and implicit *semantic augmentation (semanAug)* to address the two biases, respectively. On the one hand, we propose to address the representation bias by learning transferable and diverse representations. Specifically, we investigate the feature representations in incremental learning based on spectral analysis and present a simple technique called classAug, to let the model see more classes during training for learning representations transferable across classes. On the other hand, to overcome the classifier bias, semanAug implicitly involves the simultaneous generating of an infinite number of instances of old classes in the deep feature space, which poses tighter constraints to maintain the decision boundary of previously learned classes. Without storing any old samples, our method can perform comparably with representative data replay based approaches.

## 1 Introduction

Deep neural networks (DNNs) have enabled great success in many machine learning tasks, based on stationary, large-scale, computationally expensive, and memory-intensive training data [1, 2, 3]. Yet the need of the ability to acquire sequential experience in dynamic and open environments [4, 5, 6] poses a serious challenge to modern deep learning systems, which only perform well on homogenized, balanced, and shuffled data [7]. Typically, DNNs suffer from drastic performance degradation of previously learned tasks after learning new knowledge, which is a well-documented phenomenon, known as catastrophic forgetting [8, 9, 10]. Recently, incremental learning (IL), also referred to as lifelong learning or continual learning, has received extensive attention [11, 12, 13, 14] to enable DNNs to preserve and extend knowledge continually.

Many earlier studies focus on task-incremental learning, which uses separate output layers for different tasks, and needs the task identity for inference [11, 15, 16]. In this work, we consider a more realistic and challenging setting of class-incremental learning (Class-IL), where the model only has access to data of new classes at each stage and needs to learn a unified classifier that can classify all seen classes [13, 17, 18]. Unfortunately, the learning paradigm of Class-IL will lead to two problems: *representation bias* and *classifier bias*, as shown in Figure 1. First, for representation learning, if the feature extractor is fixed after learning old classes, the learned representations could be preserved, but suffer from the lack of transferability for new classes; on the contrary, if we update the feature extractor on new classes, the updated representations would be no longer suitable for old classes. Consequently, the old and new classes would be easily overlapped in the deep feature space. We

---

*Corresponding author.

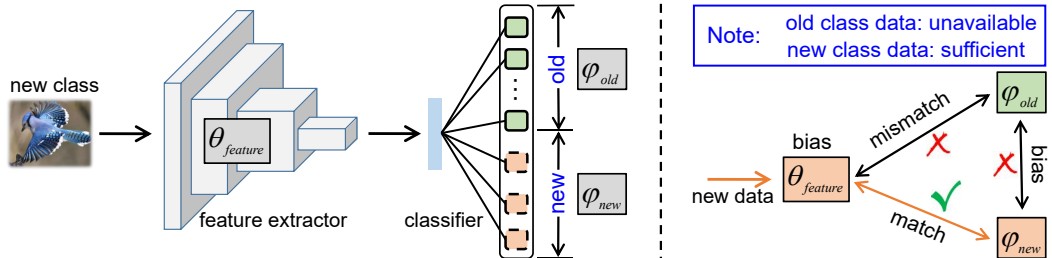

Figure 1: Two inherent problems in Class-IL: representation bias and classifier bias.

denote this dilemma as the **representation bias**. Second, to distinguish new classes from old classes, the training loss is typically calculated on all classes. Without old training data, the class weights of old classes would be ill-updated and mismatched with the updated representation space. We denote this dilemma as the **classifier bias**. In this work, we investigate the learning of representation and classifier in incremental learning and propose a simple and effective dual augmentation framework to overcome these two biases in Class-IL without storing and replaying training data of old classes.

**Learning Representation for Incremental Learning.** Existing works typically regularize network parameters explicitly [11, 15, 16] or implicitly [12] to reduce the representation shift when learning new classes. In this paper, instead of asking how to keep previously learned representations unchanged, we investigate the following question:

*What properties of learned representations could facilitate incremental learning?*

We hypothesize that learning transferable and diverse representations is an important requirement for incremental learning. Intuitively, with such representations, it could be easier to find a model to perform well on all tasks and improve both plasticity and stability, since different tasks would be closer in the parameters space. From a spectral analysis viewpoint, we investigate which components of feature representations are more transferable and less forgettable in the incremental learning process. It is found that spectral components with large eigenvalues are less forgettable. Furthermore, we exploit this finding to propose a simple technique named *classAug*, which can enlarge the spectral components to introduce more diverse and transferable representations for incremental learning.

**Learning Classifier for Incremental Learning.** Recently, several works were proposed to alleviate the classifier bias in data replay based methods [18, 19, 20]. However, in non-exemplar based (i.e., without storing and replaying old data) Class-IL setting, the classifier bias is more serious and the above methods can not be directly used. A straightforward way is storing instances of old classes in the deep feature space. However, this strategy is undesirable due to the limited memory resource and scalability. This work delves into the classifier learning for Class-IL and proposes an implicit semantic augmentation (*semanAug*) approach to generate an infinite number of instances of old classes in the deep feature space by leveraging the distribution information. SemanAug is inspired by MCF [21] and ISDA [22], which have performed semantic augmentation for linear models and DNNs, respectively. However, both our way to leverage semantic augmentation and the motivation fundamentally differ from them [21, 22].

**Contributions.** **(i)** We provide new insights into the representation learning in incremental learning by analyzing the structural characteristics of the learned embedding space via spectral decomposition and find that spectral components with large eigenvalues are less forgettable and carry more transferable features. Based on this observation, we propose a simple and effective method of classAug to learn better embedding space for incremental learning. **(ii)** For classifier learning in incremental learning, we propose semanAug which implicitly involves simultaneous generating an infinite number of instances of old classes in the deep feature space to maintain the decision boundary of previously learned classes. **(iii)** Extensive experiments on benchmark datasets demonstrate the superior performance of our dual augmentation framework for the challenging scenario of Class-IL.

## 2   Related Work

**Incremental Learning.** Diverse approaches have been proposed for incremental learning of DNNs. They can be roughly divided into three categories: regularization based, data replay based, and architecture based approaches. *Regularization based methods* focus on weight regularization by

estimating and preventing the important network weights from changing [11, 15, 16]. The difference among those methods is the way to compute the importance of the parameters. However, it is hard to design a reasonable metric to measure the importance of parameters, and it is known that regularization strategies show poor performance in Class-IL scenario [23, 24]. *Data replay based methods* address both the representation bias and classifier bias straightforwardly by storing a fraction of old data to jointly train the model with current data. With stored real samples, some works [17, 13, 25] use a distillation loss to prevent forgetting, while others [26, 27, 28] develop gradient-based regularization to make more efficient use of the rehearsal data. To avoid storing real data, another line of works generates pseudo-samples of all previous classes for replay using deep generative models [29, 30, 31, 32]. Nevertheless, storing real data is undesirable for resource-limited or privacy and safety concerning scenarios. Moreover, training big generative models for complex datasets is inefficient. *Architecture based methods* dynamically extend the network structure during the course of incremental learning [33, 34, 35, 36]. However, growing architecture is unfeasible for large numbers of tasks, and those methods are often impractical for Class-IL.

**Data Augmentation.** Literature is rich on data augmentation for improving the generalization of DNNs. Classical strategies commonly synthesize "positive" new samples in a way that is consistent with the underlying data distribution of the original dataset [3]. Recent works show that label mixing based methods such as Mixup [37] and Cutmix [38] can greatly improve the generalization of DNNs. In complement to the *input space augmentations* mentioned above, some works have explored *feature space augmentations* which augment the learned representations in deep embedding space to enhance classifier performance. The intuition behind those works is that certain directions in the deep feature space correspond to meaningful semantic transformations [39, 40]. For instance, deep feature interpolation [40] leverages simple interpolations in the embedding space to achieve semantic augmentation. A recently proposed ISDA [22] performs semantic augmentation by estimating and leveraging the category-wise distribution of deep representations in an online manner. Despite the simplicity, ISDA shows its effectiveness in semi-supervised learning [22], contrastive learning [41], domain adaptation [42] and long-tailed recognition [43].

## 3   Dual Augmentation Framework for Class-Incremental Learning

We first formalize the problem of Class-IL, and then introduce the proposed classAug for representation learning and semanAug for classifier learning, respectively. Finally, we present the dual augmentation framework for Class-IL by combing the two augmentations.

**Problem Definition.** Typically, a Class-IL problem involves the sequential learning of $\mathcal{T}$ tasks that consist of disjoint classes sets, and the model has to classify all seen classes at any given point in training. At incremental step $t \in \{1, ..., \mathcal{T}\}$, $(\boldsymbol{x}, y) \in \mathcal{D}_t$ denotes a training sample, where $\boldsymbol{x}$ is an sample in the input space $\mathcal{X}$ and $y \in \mathcal{C}_t$ is its corresponding label. $\mathcal{C}_t$ is the class set of task $t$. To facilitate analysis, we represent the DNN based model with two components: a feature extractor and a unified classifier. Specifically, the feature extractor $f_{\boldsymbol{\theta}} : \mathcal{X} \to \mathcal{Z}$, parameterized by $\boldsymbol{\theta}$, maps the input $\boldsymbol{x}$ into a feature vector $\boldsymbol{z} = f_{\boldsymbol{\theta}}(\boldsymbol{x}) \in \mathbb{R}^d$ in the deep feature space $\mathcal{Z}$; the unified classifier $g_{\boldsymbol{\varphi}} : \mathcal{Z} \to \mathbb{R}^{\mathcal{C}_{1:t}}$, parameterized by $\boldsymbol{\varphi}$, produces a probability distribution $g_{\boldsymbol{\varphi}}(\boldsymbol{z})$ as the prediction for $\boldsymbol{x}$. Denote the overall parameters by $\boldsymbol{\Theta} = (\boldsymbol{\theta}, \boldsymbol{\varphi})$.

The general objective is to correctly classify test examples from all seen classes [44]. The key challenge of Class-IL is that data from previous tasks are assumed to be unavailable, which means that the best configuration of the model for all seen tasks must be sought by minimizing the predefined loss function $\mathcal{L}$ (e.g., cross-entropy) on current data $\mathcal{D}_t$:

$$\underset{\boldsymbol{\theta}, \boldsymbol{\varphi}}{\arg\min} \ \mathbb{E}_{(\boldsymbol{x}, y) \backsim \mathcal{D}_t}[\mathcal{L}(g_{\boldsymbol{\varphi}}(f_{\boldsymbol{\theta}}(\boldsymbol{x})), y)]. \tag{1}$$

A widely used strategy to preserve old knowledge is knowledge distillation [45], which typically matches the current model with previous model response to current training data using the teacher-student framework [12, 13, 19].

### 3.1   Learning Representation with Class Augmentation

As we focus on non-exemplar based Class-IL, we intentionally avoid storing training samples of old classes. To maintain the generalizability of the learned representations for old classes, existing

methods typically restrain the feature extractor from changing [11, 15, 16, 12]. However, this would lead to a trade-off between the plasticity and stability [5], and it would be hard to perform long-step incremental learning. Our high-level idea is to learn transferable and diverse representations to bridge the old and new classes in a better feature space. To delve into this problem, we want to answer two questions: (1) **Which** part of feature representations tends to be forgotten in incremental learning? (2) **How** to facilitate the representation learning for incremental learning?

### 3.1.1 Analyzing Forgetting via Spectral Decomposition

In what follows, we explore which part of feature representations tends to be forgotten and may not be transferable across different tasks in incremental learning. To this end, we propose to quantify the sensitivity of the model to different directions in the deep feature space by measuring the similarity of the space before and after learning new tasks.

Formally, given a feature extractor $f_{\theta,old}$ trained on dataset $\mathcal{D}_{old} = \{(\boldsymbol{x}_i, y_i)\}_{i=1}^n$. A new dataset $\mathcal{D}_{new}$ that contains disjoint classes with $\mathcal{D}_{old}$ is used to update $f_{\theta,old}$, and the updated feature extractor is denoted as $f_{\theta,new}$. For the samples in $\mathcal{D}_{old}$, we can get two groups of deep features mapped by $f_{\theta,old}$ and $f_{\theta,new}$, respectively. Using eigenvalue decomposition, we could respectively decompose *the features mapped by original feature extractor (i.e., $f_{\theta,old}(\boldsymbol{x}_i)$)* as well as *the features mapped by updated feature extractor (i.e., $f_{\theta,new}(\boldsymbol{x}_i)$)* to different directions as following:

$$\frac{1}{n}\sum_{i=1}^n f_{\theta}(\boldsymbol{x}_i)f_{\theta}(\boldsymbol{x}_i)^{\mathrm{T}} = \sum_{j=1}^d \boldsymbol{u}_j \lambda_j \boldsymbol{u}_j^{\mathrm{T}}, \tag{2}$$

where $\lambda_j$ represents the eigenvalue with index $j$ and $\boldsymbol{u}_j$ is its eigenvector. $d$ is the dimensionality of the feature space. Through spectral factorization in Eq. (2), we can represent the original and new representations with two groups of eigenvectors: $\{\boldsymbol{u}_{old,1}, ..., \boldsymbol{u}_{old,d}\}$ and $\{\boldsymbol{u}_{new,1}, ..., \boldsymbol{u}_{new,d}\}$.

Next, we investigate the forgetting or transferability of each direction. Shonkwiler [46] introduced the principal angles [47] to measure the similarity of two subspaces. However, it is unreasonable to treat all eigenvectors equally to calculate the principal angles, regardless of their relative eigenvalues. Inspired by [48], we use corresponding angles, denoted by $\psi$, to explore the distance between two subspaces in incremental learning:

**Definition** 1 (**Corresponding Angle**) Given two groups of eigenvectors: $\{\boldsymbol{u}_{old,1}, ..., \boldsymbol{u}_{old,d}\}$ and $\{\boldsymbol{u}_{new,1}, ..., \boldsymbol{u}_{new,d}\}$, corresponding angle represents the angle between two eigenvectors corresponding to the same eigenvalue value index. The cosine value of the corresponding angle is:

$$\cos(\psi_j) = \frac{\langle \boldsymbol{u}_{old,j}, \boldsymbol{u}_{new,j} \rangle}{\|\boldsymbol{u}_{old,j}\| \cdot \|\boldsymbol{u}_{new,j}\|}, \tag{3}$$

where $\boldsymbol{u}_{old,j}$ is the $j$-th eigenvectors with the $j$-th largest eigenvalue in the old feature space, and similarly for $\boldsymbol{u}_{new,j}$. Note that $\|\boldsymbol{u}_{old,j}\| = 1$ and $\|\boldsymbol{u}_{new,j}\| = 1$. For IL, the meaning of "preserve old knowledge" refers to maintain the previously learned decision boundary among classes. At representation level, for an old class, the shape (i.e., covariance) of the distributions should not be changed too much. If an eigenvector direction only changes slightly after updating the feature extractor, the corresponding angle is small, and vice versa. Intuitively, the corresponding angle could capture the representation shift between the old and updated feature extractor during incremental learning, and reflect the forgetting along certain directions in the deep feature space.

Based on the metric defined above, we explore the forgetting of different directions in Class-IL. We use LwF-MC [12, 13] as baseline method and train a ResNet-18 [1] on CIFAR-100 [49] using SGD in a 2-step manner. Concretely, the model is first trained on the first 50 classes and then updated on the other 50 classes. Figure 2 (a) shows the absolute cosine values of corresponding angles between the old and new eigenvectors. We can observe that eigenvectors with larger eigenvalues produce larger similarity (small corresponding angles), which indicates those directions are more transferable and less forgettable across different tasks. On the contrary, the eigenvectors with small eigenvalues prefer to move after updating the model on new tasks, and could be regarded as forgettable directions.

**Transferable and Diverse Representations.** As demonstrated above, the directions with larger eigenvalues transfer better and suffer less forgetting. This thought-provoking observation indicates that our learned representations should have the following properties: **(1) Transferability**: the eigenvalues of those several significant directions should be enlarged to transfer across tasks (or

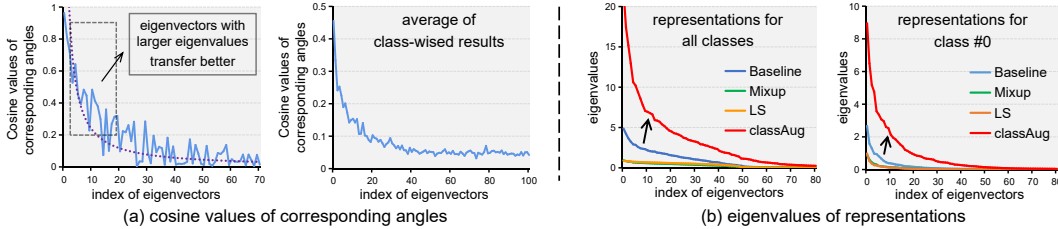

Figure 2: (a) Absolute cosine values of corresponding angles. (b) Distribution of eigenvalues for baseline, Mixup [37], LS [50], and our classAug training based models.

classes). **(2) Diversity**: the number of the directions with significant eigenvalues should be increased. Note that those properties are different from that in the common single-task learning scenario. Actually, reducing the number of directions with significant variance has been seen as a form of *feature compression* [51], which is linked to generalization by information theory [52, 53]. However, the usual concepts of generalization may not entirely be appropriate for IL, since standard learning only aims to learn compact representations within training classes without considering new class generalizability. In IL, those less discriminative directions for the current task could capture useful representations for future tasks. A recent paper [54] has shown that strong compressed representations can actually hurt the generalization ability in the deep metric learning setting. Therefore, to reduce forgetting and enhance the transferability of the representations, it is important to enlarge the eigenvalues and increase the number of eigenvectors with significant variance.

### 3.1.2 Learning Representations via Class Augmentation

We now exploit our above analysis to propose a simple method for representation learning in Class-IL. Our key idea is to learn transferable and diverse representations by learning more classes at each incremental stage $t$. To do so, a direct way is to introduce real classes from other datasets as auxiliary. However, it is unrealistic to always have access to other real classes, and which datasets should be used remains unknown. Therefore, we propose **class augmentation (classAug)** to augment the original classes by synthesizing auxiliary classes based on $\mathcal{D}_t$. Concretely, inspired by Mixup [37], classAug randomly interpolates two samples $\boldsymbol{x}_a$ and $\boldsymbol{x}_b$ from two different classes $a$ and $b$ to generate a new sample $\boldsymbol{x}_{ab}^{\text{new}}$ representing a new class:

$$\boldsymbol{x}_{ab}^{\text{new}} = \lambda \boldsymbol{x}_a + (1 - \lambda)\boldsymbol{x}_b, \qquad (4)$$

where $\lambda$ is a *random* number of interpolation coefficient. For a $k$-class problem, we can generate $k(k-1)/2$ new classes using the above method, which can be further merged to $m$ auxiliary classes. As a result, the original $k$-class

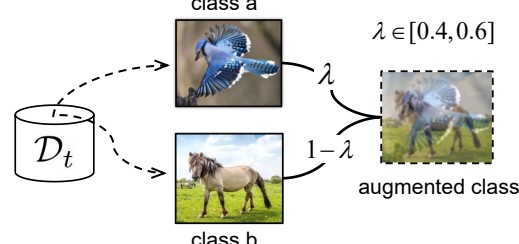

Figure 3: Illustration of classAug.

problem in the current task is extended to a $(k+m)$-class problem. Moreover, we restrict the $\lambda$ to be sampled from the interval of $[0.4, 0.6]$, to reduce the overlap between the augmented and original classes. At the end of each IL stage, the augmented class nodes in the classifier would be removed.

**Discussion.** The proposed classAug is related to Mixup [37] which applies random interpolation on a pair of training samples and the respective one-hot labels. However, the interpolated samples in Mixup are near original data, and the number of classes is not changed, but in our method, it is increased. By learning to classify more classes in each stage $t$, the model could learn more transferable and diverse representations. Figure 2 (b) displays and compares the eigenvalues [2] of representations learned with different methods on the first 50 classes of CIFAR-100. It is obvious that the proposed classAug can enhance the value of eigenvalues significantly, and produce more directions with significant variance compared with other methods. On the contrary, Mixup and Label-Smoothing (LS) [50] lead to significantly smaller eigenvalues for the several top eigenvectors, which represent more compact representations. Indeed, the compression effect of soft-label based methods has also been demonstrated in [51, 50]. As shown in Section 4.3, classAug can improve the performance of Class-IL significantly, while Mixup and LS have negative effect in our experiments.

---

[2]To visualize the distribution clearly, we do not include the largest eigenvalue in the figure.

## 3.2 Learning Classifier with Semantic Augmentation

As demonstrated in Section 1, classifier bias is another problem in Class-IL. When learning new classes, the previously learned decision boundary would suffer from catastrophic distortion and thus the test samples from old classes could be easily mapped to wrong classes. To overcome this issue, we propose **semantic augmentation (semanAug)**, which leverages the distribution information (i.e., class mean and covariance) of old classes to regularize the learning of the classifier. Formally, for each old class $k \in \{1, ..., \mathcal{C}_{old}\}$, we can generate $M$ instances in the deep feature space from its distribution, i.e., $\widetilde{z}_k \backsim \mathcal{N}(\boldsymbol{\mu}_k, \gamma \boldsymbol{\Sigma}_k)$, in which $\gamma$ is a non-negative coefficient. Then the generated instances of old classes and real instances of new classes in the deep feature space can be jointly fed to the classifier for minimizing cross-entropy loss:

$$\mathcal{L}_t = \underbrace{\frac{1}{n_t} \sum_{i=1}^{n_t} -\log\left( \frac{e^{\boldsymbol{\varphi}_{y_i}^{\mathrm{T}} \boldsymbol{z}_i + b_{y_i}}}{\sum_{c=1}^{\mathcal{C}_{all}} e^{\boldsymbol{\varphi}_c^{\mathrm{T}} \boldsymbol{z}_i + b_c}} \right)}_{\mathcal{L}_{t,new}: \text{ loss on real features of new classes}} + \underbrace{\frac{1}{\mathcal{C}_{old}} \sum_{k=1}^{\mathcal{C}_{old}} \frac{1}{M} \sum_{m=1}^{M} -\log\left( \frac{e^{\boldsymbol{\varphi}_k^{\mathrm{T}} \widetilde{\boldsymbol{z}}_{k,m} + b_k}}{\sum_{c=1}^{\mathcal{C}_{all}} e^{\boldsymbol{\varphi}_c^{\mathrm{T}} \widetilde{\boldsymbol{z}}_{k,m} + b_c}} \right)}_{\mathcal{L}_{t,old}: \text{ loss on generated features of old classes}}, \quad (5)$$

where $n_t$ is the number of training samples in current task dataset $\mathcal{D}_t$, $\mathcal{C}_{old}$ is the number of total old classes upon stage $t$, and $\mathcal{C}_{all} = \mathcal{C}_{old} + \mathcal{C}_t$ is the number of all seen classes at stage $t$. $\boldsymbol{\varphi} = [\boldsymbol{\varphi}_1, ..., \boldsymbol{\varphi}_{\mathcal{C}_{all}}]^{\mathrm{T}} \in \mathcal{R}^{\mathcal{C}_{all} \times d}$ and $b = [b_1, ..., b_{\mathcal{C}_{all}}]^{\mathrm{T}} \in \mathcal{R}^{\mathcal{C}_{all}}$ are the weight matrix and bias vector of the last fully connected layer, respectively.

In Class-IL, the second term in Eq. (5), $\mathcal{L}_{t,old}$, is computationally inefficient when $M$ and $\mathcal{C}_{old}$ are large. In the following, we present an easy-to-compute way to implicitly generate infinite instances in the deep feature space for old classes.

**Upper bound of $\mathcal{L}_{t,old}$.** Concretely, in the case of $M \rightarrow \infty$, the second term in Eq. (5):

$$\begin{aligned} \mathcal{L}_{t,old} &= \frac{1}{\mathcal{C}_{old}} \sum_{k=1}^{\mathcal{C}_{old}} \mathbb{E}_{\widetilde{\boldsymbol{z}}_k} \left[ -\log\left( \frac{e^{\boldsymbol{\varphi}_k^{\mathrm{T}} \widetilde{\boldsymbol{z}}_k + b_k}}{\sum_{c=1}^{\mathcal{C}_{all}} e^{\boldsymbol{\varphi}_c^{\mathrm{T}} \widetilde{\boldsymbol{z}}_k + b_c}} \right) \right] = \frac{1}{\mathcal{C}_{old}} \sum_{k=1}^{\mathcal{C}_{old}} \mathbb{E}_{\widetilde{\boldsymbol{z}}_k} \left[ \log\left( \sum_{c=1}^{\mathcal{C}_{all}} e^{(\boldsymbol{\varphi}_c^{\mathrm{T}} - \boldsymbol{\varphi}_k^{\mathrm{T}}) \widetilde{\boldsymbol{z}}_k + (b_c - b_k)} \right) \right] \\ &\leqslant \frac{1}{\mathcal{C}_{old}} \sum_{k=1}^{\mathcal{C}_{old}} \log\left( \mathbb{E}_{\widetilde{\boldsymbol{z}}_k} \left[ \sum_{c=1}^{\mathcal{C}_{all}} e^{(\boldsymbol{\varphi}_c^{\mathrm{T}} - \boldsymbol{\varphi}_k^{\mathrm{T}}) \widetilde{\boldsymbol{z}}_k + (b_c - b_k)} \right] \right) \\ &= \frac{1}{\mathcal{C}_{old}} \sum_{k=1}^{\mathcal{C}_{old}} \log\left( \sum_{c=1}^{\mathcal{C}_{all}} e^{\boldsymbol{v}_{c,k}^{\mathrm{T}} \boldsymbol{\mu}_k + (b_c - b_k) + \frac{\gamma}{2} \boldsymbol{v}_{c,k}^{\mathrm{T}} \boldsymbol{\Sigma}_k v_{c,k}} \right). \end{aligned}$$
$$(6)$$

In above equation, $\boldsymbol{v}_{c,k} = \boldsymbol{\varphi}_c - \boldsymbol{\varphi}_k$. The inequality is based on Jensen's inequality $\mathbb{E}[\log(X)] \leqslant \log\mathbb{E}[X]$, and the last equality is obtained by using the moment-generating function $\mathbb{E}[e^{tX}] = e^{t\mu + \frac{1}{2}\sigma^2 t^2}$, $X \backsim \mathcal{N}(\mu, \sigma^2)$, due to the fact that $(\boldsymbol{\varphi}_c - \boldsymbol{\varphi}_k)\widetilde{\boldsymbol{z}}_k + (b_c - b_k)$ is a Gaussian random variable. As can be seen, Eq. (6) is an upper bound of original $\mathcal{L}_{t,old}$, which provides an elegant and much efficient way to **implicitly** generate **infinite** instances in the deep feature space for old classes. The $\mathcal{L}_{t,old}$ in Eq. (6) can be write in the common cross-entropy loss form:

$$\mathcal{L}_{t,semanAug} \triangleq \mathcal{L}_{t,old} = \frac{1}{\mathcal{C}_{old}} \sum_{k=1}^{\mathcal{C}_{old}} -\log\left( \frac{e^{\boldsymbol{\varphi}_k^{\mathrm{T}} \boldsymbol{\mu}_k + b_k}}{\sum_{c=1}^{\mathcal{C}_{all}} e^{\boldsymbol{\varphi}_c^{\mathrm{T}} \boldsymbol{\mu}_k + b_c + \frac{\gamma}{2} \boldsymbol{v}_{c,k}^{\mathrm{T}} \boldsymbol{\Sigma}_k v_{c,k}}} \right). \quad (7)$$

Intuitively, $\mathcal{L}_{t,old}$ implicitly performs semantic transformations for $\boldsymbol{\mu}_k$ based on $\boldsymbol{\Sigma}_k$. To maintain the decision boundary, $\gamma$ should be smaller if the distribution of a class is near the decision boundary; instead, $\gamma$ should be bigger if the distance is relatively far. We set $\gamma = 2$ in our experiments. In addition, we can observe that when $\gamma = 0$, only the class means are used for knowledge retention.

**Discussion. (1)** Although the derivation of the upper bound in Eq. (6) is similar with ISDA [22], both our motivation and the way to leverage semanAug are different from ISDA. When learning new classes, we only *apply semanAug for the class mean of each old class* based on the memorized distribution information. While ISDA applies semanAug on all the training samples to improve generalization in standard supervised learning. In addition, a crucial step in ISDA is to estimate the mean and covariance matrix of each class in an online manner. Differently, semanAug is naturally suitable for Class-IL, since the distribution of old classes can be estimated with all training samples at the end of each learning stage. **(2)** Using previous class statistics for IL has also been explored in IL2M [55]. However, our method differs from IL2M in both the statistics information and the way to

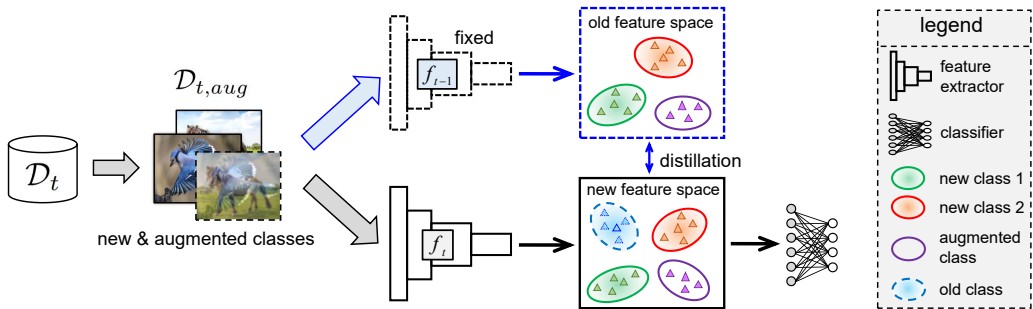

Figure 4: Illustration of our dual augmentation framework (IL2A) for Class-IL. On the one hand, the training samples of new classes at current task are augmented via the proposed classAug. On the other hand, the distributions of old classes are retained by semanAug in the deep feature space.

leverage them. First, The class statistics in IL2M is the prediction score of the classifier, while ours is the class distribution statistics in the deep feature space. Second, IL2M uses the class statistics to calibrate the prediction of a continual learner in a post-processing manner, while our method leverage the statistics to automatically learn a balanced classifier.

### 3.3 The Dual Augmentation Learning Framework

With classAug for representation bias and semanAug for classifier bias, Figure 4 describes the learning process of the dual augmentation framework (**IL2A**). We also use the well-known knowledge distillation (KD) [19] for two reasons. Firstly, classAug and KD are complementary and focus on different aspect of learning representation. Secondly, KD can reduce the change of feature extractor, which is crucial for semanAug because it implicitly generate instances in the deep feature space from old distribution. The total learning objective at each stage $t$ is as following:

$$\mathcal{L}_t = \mathcal{L}_{t,new} + \alpha \mathcal{L}_{t,semanAug} + \beta \mathcal{L}_{t,kd}, \tag{8}$$

where $\alpha$ and $\beta$ are two hyper-parameters. $\mathcal{L}_{t,new}$ and $\mathcal{L}_{t,semanAug}$ are shown in Eq. (5) and Eq. (7), respectively. $\mathcal{L}_{t,kd} = \frac{1}{n_t} \sum_{i=1}^{n_t} \| f_{\theta_{t-1}}(x_i) - f_{\theta_t}(x_i) \|$. Note that $\mathcal{L}_{t,new}$ and $\mathcal{L}_{t,semanAug}$ are applied to both the original and synthesized samples. Algorithm 1 presents the pseudo code of IL2A.

## 4 Experiments

### 4.1 Evaluation Protocol

**Datasets.** We perform our experiments on CIFAR-100 [49] and Tiny-ImageNet [56]. A common setting is to train the model on half of classes for first task, and equal classes in the remaining incremental steps. Based on this, we split the CIFAR-100 dataset in different settings: $50 + \mathbf{5} \times 10$, $50 + \mathbf{10} \times 5$, $40 + \mathbf{20} \times 3$. For instance, $50 + \mathbf{10} \times 5$ represents that the first task contains 50 classes and there are 5 classes for the following 10 tasks. Similarly, the settings for Tiny-ImageNet are $100 + \mathbf{5} \times 20$, $100 + \mathbf{10} \times 10$ and $100 + \mathbf{20} \times 5$. Intuitively, more classes in each tasks requires the model to learn a harder problem for each task, while increasing the length of the task sequence challenges the model's retention.

---

**Algorithm 1:** IL2A: Dual augmentation algorithm

Randomly initialize $\Theta^0 = \{\theta^0, \varphi^0\}$; $\mathcal{S}^0 = \emptyset$;
**foreach** *incremental stage* $t \in \{1, ..., \mathcal{T}\}$ **do**
  **Input:** model $\Theta^{t-1}$, data $\mathcal{D}_t = \{(x_i, y_i)\}_{i=1}^{n_t}$;
  **Output:** model $\Theta^t$;
  $\Theta^t \leftarrow \Theta^{t-1}$;
  $\mathcal{D}_{t,aug} = \{(x_i', y_i')\}_{i=1}^{n_t'}$ via classAug;
  add class nodes for augmented classes;
  **if** $t = 1$ **then**
    | train $\Theta^t$ by minimizing $\mathcal{L}(g_\varphi(f_\theta(x')), y')$;
  **else**
    | train $\Theta^t$ by minimizing Eq. (8);
  $s \leftarrow$ compute $\{\mu, \Sigma\}$ for each class in $\mathcal{D}_t$;
  $\mathcal{S}^t \leftarrow \mathcal{S}^{t-1} \cup s$;
  remove augmented class nodes in classifier;

---

**Implementation Details.** In our experiments, we follow [44] to utilize the ResNet-18 [1] as our base architecture, and train it from scratch

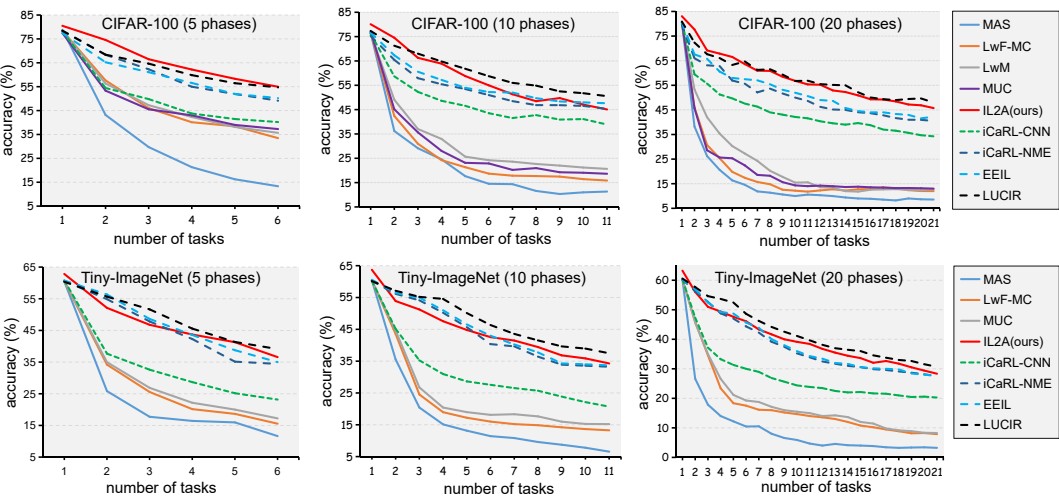

Figure 5: Results of top-1 accuracy on CIFAR-100 and Tiny-ImageNet under different settings. Solid lines present methods that do not store old exemplars, dashed lines present data replay based methods.

in each experiment. All models are trained using Adam [57] optimizer with an initial learning rate of 0.001 for 100 epochs with the mini-batch size of 64. The learning rate is reduced by a factor of 10 at 45 and 90 epochs. We use the same hyper-parameter value for all experiments. Specifically, we set $\alpha = 10$ and $\beta = 10$ in Eq. (8). The number of augmented classes (i.e. The number of augmented classes (i.e., $m$) depends on the number of (original) classes at current incremental step. Taking CIFAR-100 as an example, the $m$ is 45 for 5 phases setting where each incremental step has 10 classes; and $m$ is 10 for 10 phases setting where each incremental step has 5 classes. At the end of each incremental stage, we evaluate the model on all seen classes after removing the class nodes of the $m$ augmented classes in the classifier. Our code is available at https://github.com/Impression2805/IL2A.

**Comparison Methods.** Our method (IL2A) does not store any old samples for replay when learning new classes. Therefore, we first compare IL2A with several non-exemplar based approaches: MAS [16], LwF-MC [13], MUC [58], LwM [59]. In addition, we also compare with several exemplar based methods such as iCaRL [13], EEIL [18] and LUCIR [19]. Specifically, for the data replay based methods, we follow [13, 19] to store 20 samples for each class using 'herd' selection technique [13]. We report the average top-1 accuracy of all previously seen classes up to each incremental step $t$. For iCaRL, we respectively report its results of CNN predictions and nearest-mean-of-exemplars classification, denoted as iCaRL-CNN and iCaRL-NME.

## 4.2 Experimental Results

**Main Results.** Comparative results are shown in Figure 5. Firstly, we observe that our method performs much better than non-exemplar based methods such as LwF-MC and MUC in the trend of accuracy curve under different settings. Particularly, the gap appears unbridgeable in the long-step Class-IL setting, e.g., 10 phases and 20 phases. This suggests that only constraining old parameters does not suffice to prevent forgetting. We argue that this is partly due to the unaddressed classifier bias. When compared to representative data replay based methods such as iCaRL, EEIL and LUCIR, our method remarkably shows strong performance without storing old samples.

The success of our method can contribute to the proposed classAug and semanAug. Specifically, classAug is applied to new classes of current task, which enables the model to learn more transferable and diverse representations for future classes and in turn, reduces the forgetting of old parameters when learning new classes. While semanAug is applied to old classes of previous tasks, which leverage the valuable distribution information of old classes to learn a unified classifier to connect the classes from different tasks to each other.

**Ablation Study.** To evaluate the effect of each component in IL2A, we perform the ablation study and show the results of 10 phases setting (CIFAR-100) in Table 1. Specifically, the **baseline** denotes the method that does not generate pseudo-instance using semanAug, but only replays the class-mean

Table 1: The effect of each component in IL2A.

| Method\Incremental stage | 1 | 2 | 3 | 4 | 5 | 6 | 7 | 8 | 9 | 10 | Final |
|---|---|---|---|---|---|---|---|---|---|---|---|
| Knowledge Distillation | 78.78 | 30.18 | 20.71 | 14.61 | 11.87 | 8.80 | 7.70 | 7.23 | 7.10 | 6.05 | 6.04 |
| baseline | 78.86 | 62.85 | 56.96 | 54.66 | 51.72 | 47.33 | 43.61 | 40.12 | 40.76 | 36.55 | 34.71 |
| + semanAug | 79.16 | 69.14 | 60.68 | 58.18 | 54.77 | 50.89 | 48.45 | 46.29 | 46.97 | 44.38 | 42.09 |
| + classAug | 79.72 | 68.30 | 64.15 | 60.15 | 56.21 | 52.61 | 51.48 | 46.48 | 46.36 | 43.63 | 41.56 |
| + classAug + semanAug | **81.08** | **74.54** | **66.28** | **63.89** | **58.80** | **54.97** | **51.32** | **48.64** | **49.74** | **47.05** | **45.07** |

of each old class when training new classes. By doing so, we aim to validate the effectiveness of semanAug compared with only replaying class-mean. In summary, we can observe that: **(1)** Baseline improves the performance of KD significantly. **(2)** SemanAug improves the performance of baseline from $34.71\%$ to $42.09\%$. Those results indicate the effect of the distribution information for maintaining old knowledge in Class-IL. **(3)** ClassAug also has remarkably effect on baseline, and **(4)** the performance can be further improved by combing with semanAug, which indicates that those two modules are complementary. Similar results are observed in other settings of CIFAR-100 and Tiny-ImageNet datasets. **(5)** As for the computational complexity, classAug involves input level sample mixing and the augmented samples are fed to feature extractor. Differently, semanAug performs implicit old instance generation in the deep feature space. Therefore, semanAug is cheaper compared with classAug from the computation perspective.

### 4.3 Further Analysis

**ClassAug Improves both Plasticity and Stability in Class-IL.** To analyze the effectiveness of classAug more concretely, we explore how it affects the new tasks accuracy ($\uparrow$) and average forgetting ($\downarrow$) (CIFAR-100, 10 phases setting). Average forgetting [60] is defined to estimate the forgetting of previous tasks. The forgetting measure $f_k^i$ of the $i$-th task after training $k$-th task is defined as $f_k^i = \max_{t \in 1,...,k-1}(a_{t,i} - a_{k,i}), \forall i < k$, in which $a_{m,n}$ is the accuracy of task $n$ after training task $m$. The average forgetting measure $F_k$ is then defined as $F_k = \frac{1}{k-1}\sum_{i=1}^{k-1} f_k^i$. Intuitively, new task accuracy can be viewed as the plasticity of the incremental learner and the average forgetting can be viewed as the stability of the incremental learner. Figure 6 (a) and (b) report the results, from which we see that classAug simultaneously improves the new task accuracy and reduces the average forgetting. Specifically, the significant improvement on new task accuracy implies that the model training with classAug is a good initialization for the following tasks. Consequently, classAug is effective to improve the trade-off between plasticity and stability of a continual learner.

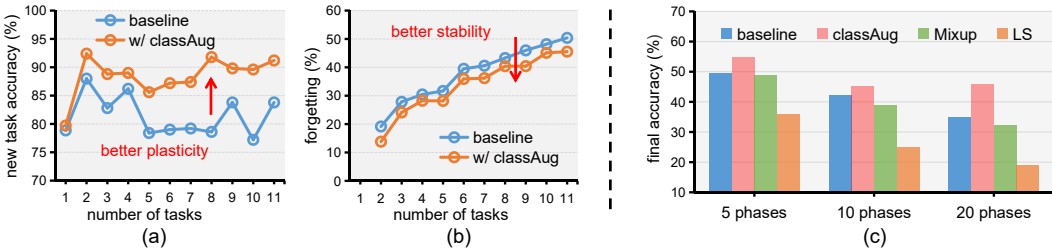

Figure 6: (a, b) ClassAug can simultaneously improve the new task accuracy and reduce the average forgetting. (c) Compared with classAug, Mixup and LS have negative effect for Class-IL.

**Compare ClassAug with Other Regularizers.** We compare the proposed classAug with Mixup and LS in Figure 6 (c), where the baseline (with semanAug) represents our IL2A without using classAug. As can be seen, Mixup and LS have negative effect on the final accuracy. This phenomenon could be interpreted based on the analysis in Section 3.1.1 and Figure 2 (b). Specifically, those regularizers result in more compressed representations, damaging the transferability of the representations. Besides, the label smoothing strategy also affects the weights of old classes in the classifier, thus increasing the classifier bias. Similar results have also been reported in [61].

**Discussion of Covariance Matrix.** In our main experiments, we use the original covariance matrix for semanAug. However, storing the original covariance matrix might be inefficient when the

Table 2: OOD detection results. ↑ indicates higher is better.

| OOD | AUROC ↑ | | | AUPR-In ↑ | | | AUPR-Out ↑ | | |
|---|---|---|---|---|---|---|---|---|---|
| | baseline | Mixup | classAug | baseline | Mixup | classAug | baseline | Mixup | classAug |
| MNIST | 87.02 | 92.46 | **94.99** | 79.89 | 89.00 | **93.05** | 92.26 | 95.48 | **97.20** |
| Fashion-MNIST | 90.28 | 93.37 | **94.40** | 86.18 | 89.11 | **92.43** | 94.26 | 96.19 | **96.78** |
| LSUN | 88.50 | 88.80 | **93.90** | 83.48 | 74.71 | **91.08** | 92.92 | 94.09 | **96.73** |
| Tiny-ImageNet | 88.49 | 84.96 | **93.92** | 83.84 | 64.02 | **91.77** | 92.70 | 92.19 | **96.55** |
| Mean | 88.57 | 89.90 | **94.30** | 83.35 | 79.21 | **92.08** | 93.04 | 94.49 | **96.81** |

matrix dimension is large. An alternative way is to only store the elements on the diagonal, which could greatly reduce the cost of memory. Figure 7 also reports the results of using the diagonal covariance matrix. Under different settings, using the original covariance matrix is slightly better than the diagonal form. This is reasonable because the original covariance matrix stores more distribution information of old classes. However, using the diagonal covariance matrix would be more memory-efficient in practice.

**ClassAug Improves Confidence Reliability.** During continuous use of a machine learning system in open-world applications, there are mainly three key steps [62]. The first step is out-of-distribution (OOD) detection [63], which requires the system to detect unknown samples from novel classes. The second step is to label the collected unknown samples by humans or automatic algorithms [64]. Finally, the system must scale and adapt incrementally to learn the novel classes, which is the Class-IL problem studied in this paper. Recently studies found that DNNs are overconfident for their predictions [63, 65], lacking the ability to detect samples from unknown classes. In real-world applications, we expect a continual learner has good OOD detection ability.

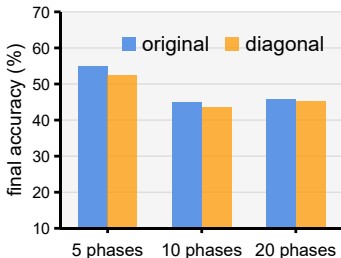

Figure 7: Original v.s. diagonal covariance matrix. CIFAR-100.

We explore the OOD detection ability of the proposed classAug. Concretely, we train a ResNet-18 on CIFAR-10, and the test samples from CIFAR-10 are in-distribution. For OOD examples, we test on MNIST [66], Fashion-MNIST [67], LSUN (resized) [68] and Tiny-ImageNet (resized). As shown in Table 2, classAug noticeably improves the OOD detection performance of baseline [63] on commonly used metrics such as AUROC, AUPR-In and AUPR-Out [63]. By recognizing synthetic samples, DNNs could learn more robust and transferable representations which could be generalized to OOD samples. Moreover, as shown in Table 2, Mixup sometimes damages the performance of OOD detection, which further demonstrates the superiority of classAug.

# 5 Conclusion

In this paper, we propose a simple and effective dual augmentation framework to address the representation bias and classifier bias in Class-IL. We first investigate the transferability (or forgetting) of representations via spectral decomposition, which motivates us to propose classAug that can learn transferable, diverse and less compact representations for IL. Furthermore, we propose to use semanAug to implicitly generate infinite instances of old classes in the deep feature space during jointly learning of the unified classifier. Experiments show that our method could achieve remarkable performance compared with state-of-the-art Class-IL methods. Future works will consider the dual augmentation framework for more challenging scenarios like Class-IL with distribution shift and OOD data, few-shot Class-IL, and federated incremental learning.

# Acknowledgements

This work has been supported by the National Key Research and Development Program under Grant No. 2018AAA0100400, the National Natural Science Foundation of China (NSFC) grants U20A20223, 61633021, 62076236, 61721004, the Key Research Program of Frontier Sciences of CAS under Grant ZDBS-LY-7004, and the Youth Innovation Promotion Association of CAS under Grant 2019141.

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
