# OpenReview forum: "Class-Incremental Learning via Dual Augmentation"
_NeurIPS.cc/2021/Conference — NeurIPS 2021 Poster_

### Official Review · Reviewer_9tuP · 2021-07-15

**Rating:** 6
**Confidence:** 4

**Summary:**

The authors propose a non-replay class-incremental learning method that works by combining input mixup with pseudo-replay at feature level. The authors motivate this method by the means of a simple study on how feature space eigenvectors change while training. The proposed approach is evaluated against a collection of current continual learning methods and shows competitive performance.


**Limitations And Societal Impact:**

This is a theoretical work in continual learning classification. This is a very abstract field that does not elicit any immediate application. As such, I feel that no discussion on potential societal impacts is necessary.

**Main Review:**


In general, I found the technical core of this work to be very interesting:
- I found semanAug to be a simple yet very interesting approach. Recent Continual Learning literature often disregards feature pseudo-rehearsal to focus instead on input replay or generation. According to the results shown by the authors, this approach is deserving of further exploration.
- The overall baseline proposed by the authors is very simple and yet manages to show remarkable performance, given its lack of a memory buffer. It is very remarkable to see it performing on par with memory-based methods, which up to now are generally regarded as the only viable solution for class-incremental learning.

However, this work is characterized by several weaknesses, that I feel need to be addressed to increase its potential:
- In the introduction, the authors make a distinction between classifier and representation bias. While the latter clearly refers to the mismatch between the features learned in past tasks vs the ones needed for new tasks, I cannot clearly understand what the authors mean by classifier bias. The definition provided at lines 32-33 seems to me a general description of the stability-plasticity dilemma, the issue at the core of continual learning and widely debated in literature [11, 17 in the paper]. I do not see this as a bias to be solved, but rather as a generally valid description of what happens when training a neural network on incremental data. Along the same line, the authors claim at line 128 that setting a trade-off between plasticity and stability hinders learning in the long run. Again, I see this as a choice that need to be made and that is not negative per se: whether it hinders training only depends on the produced amount of stability.
- The authors claim at lines 44-45 that “learning diverse and transferable representation is an important requirement […] which has been ignored by previous works”. I do not find this statement to be accurate: finding suitable and transferable representations has always been one of the cornerstones of continual learning approaches [13, 28 in the paper, Joseph et al. 2020, Chaudhry et al. 2020]
- This paper adopts an unclear position towards Replay-based methods: on the one hand, the authors claim that they are deliberately targeting non-exemplar-based continual learning, on the other hand, they only adopt replay-based method for their preliminary experiment in section 3.1 and compare IL2A with several replay methods in section 4, so they are not limiting themselves to the aforementioned setting. Even though the method proposed by the authors requires retaining a previous snapshot of the model (which amounts to a memory footprint that may be comparable with replay methods), their non-exemplar-based proposal performing on par with rehearsal approaches is a significant result. However, I wonder if their proposed approach can be successfully coupled with replay and get to even better performance. Indeed, adding such a simple additional experiment could make the whole work even more relevant.
- A continual learning work that shares some similarities with the authors’ proposal is IL2M (Belouadah et al., 2019), which also makes use of previous class statistics to rectify the prediction scores and prevent representation bias. I think that adding a direct comparison with how this principle is differently exploited in this work could provide a comprehensive picture for the readers.
- I found the experimental sections to be often unclear so I strongly suggest that the authors take further steps to improve it, with particular reference to the following points:
    - I could not understand the meaning of the numbers contained in Table 1. Are they average accuracy values or incremental accuracy values? Are these numbers referring to different experiment (50-split dataset, 60-split dataset, etc.) or do they refer to the same experiment? What dataset is being used?
    - What does “eigenvalues of covariance” mean at line 182? How exactly is the intrinsic dimension of features defined at line 188? When introducing a specific approach for evaluation, it could be useful to explicitly present it to the reader or at least introduce relevant references.
    - Running continual learning experiments is often a complex matter, since there exist different experimental settings which are often incompatible. In order to evaluate whether experiments are fairly conducted, since the authors did not submit the code for review, but state they will publish it upon acceptance, I need more details on the way experiments were conducted. What codebase was used for the experiments, a new one created from scratch or was it based on the code from other works? Are all results computed anew for all methods or taken from other works? How was hyper-parameter selection conducted for the competitors, were they tuned at all or were original hyperparameters used?
    - In the ablation study, I could not clearly understand how the baseline is constructed: how is the class-mean of old classes used to regularize the classifier (line 311)? Does this mean that only part of the semanAug loss is used? I could not understand this from the text.
    - How is the forgetting measure at line 324 defined? There are several instances of such measure in CL literature, so I would need a citation to understand which one the authors are referring to.

The following minor issues did not affect my evaluation directly, but I list them for the authors to consider:
- I believe that adding variance values for the presented results (at least in the supplementary material) would be very useful to further simplify the understanding of the presented data.
- Since the denominator in equation 3 is 1, the equation could be simplified.
- In line 202, the authors define [56] “a very nice recent paper”, I would recommend keeping a neutral tone instead.
- The comic sans typeface was used in most presented figures, which is unusual in scientific papers. A common sans-serif font would probably be more appropriate.
- There are several typos that need fixing:
Line 39: learning, and → learning and
Line 49: is benefited from → benefits from
Line 62: to storing → to store / storing
Line 96: Classical strategies commonly synthetic → verb is missing
Line 99: in complementary → in complement
Line 129: our high idea → our high-level idea
Line 136: naturally rises → naturally raises
Line 140: How the regularization affect → How does the regularization affect
Line 143: are tend to be forgetting → tend to be forgotten
Line 149: is donated as → is denoted as
Line 161: two groups eigenvectors → two groups of eigenvectors
Line 185: We note that simimlar → We note that a similar
Line 236: tast samples → task samples
Line 236: wrong class → wrong classes
Line 237: high level idea to → high-level idea is to
Caption of figure 4: hans → hand
Line 266: differnt → different
Line 274: rest incremental step → remaining incremental steps
Line 300: bais → bias
Line 306: task → tasks
Line 310: method that using → method using
Line 311: and use the class → and class

(Joseph et al., 2020) Meta-consolidation for continual learning, NeurIPS 2020
(Chaudhry et al., 2020) Continual Learning in Low-rank Orthogonal Subspaces, NeurIPS2020
(Belouadah et al., 2019) Il2m: Class incremental learning with dual memory, ICCV 2019


**Time Spent Reviewing:**

5

---

> ### Author Response · Authors · 2021-08-10
> **Response to Reviewer 9tuP**
>
> We sincerely appreciate your thoughtful comments, efforts, and time. We respond to each of your questions and concerns one-by-one in what follows:
>
> **Q1: ..., I cannot clearly understand what the authors mean by classifier bias. .., the authors claim at line 128 that setting a trade-off between plasticity and stability hinders learning in the long run. Again, I see this as a choice...**
>
> - Classifier bias. We explain the classifier bias in L35-L37, where we refer to the imbalance between the classifier weights of old and new classes. The classifier weights for old classes are over-punished when learning new classes, causing a bias between the weights of old and new classes. There are several works that address this problem for input-replay based methods (e.g., [15, 16, 21, 45]). In this paper, we aim to address this problem without storing any samples for old classes.
>
> - Plasticity-stability dilemma. We agree with your opinion about plasticity and stability. Our point is: although plasticity-stability dilemma always exists in continual learning, there might exist some ways that could improve both plasticity and stability of a continual learner, thus improving the overall performance. The results in Fig.6 confirm that the proposed method could simultaneously improve the new task accuracy and average forgetting.
>
> **Q2: The authors claim at lines 44-45 that “learning diverse and transferable..., which has been ignored by previous works”. I do not find this statement to be accurate...**
>
> Thanks for pointing out the imprecision of our description. We will modify our statement and add more discussion on “learning transferable representations” for incremental learning based on the papers you suggested.
>
> **Q3: I wonder if their proposed approach can be successfully coupled with replay and get to even better performance. Indeed, adding such a simple additional experiment could make the whole work even more relevant.**
>
> Thanks for this valuable suggestion. We have added an experiment to combine our approach with replay-based methods. The results are as follows (CIFAR-100: 50+10*5 setting: the first task includes 50 classes, and each of the other 10 tasks contains 5 classes). As can be seen, the performance of input-replay based method can be further improved when combined with our method.
>
> |**Step**|1|2|3|4|5|6|7|8|9|10|11|
> |---|---|---|---|---|---|---|---|---|---|---|---|
> |**iCaRL**|78.28|67.44|60.92|55.43|53.47|50.97|48.49|46.85|46.83|46.52|45.31|
> |**+ours**|**80.82**|**71.15**|**63.70**|**60.89**|**58.77**|**55.88**|**53.39**|**51.53**|**50.42**|**50.33**|**49.83**|
>
>
>
> **Q4: IL2M (Belouadah et al., 2019) also makes use of previous class statistics to rectify the prediction scores and prevent representation bias. I think that adding a direct comparison with how this principle is differently exploited in this work could provide a comprehensive picture for the readers.**
>
> Thanks for drawing our attention to IL2M, which not only stores old samples but also uses old class statistics. Our method differs from IL2M in both the statistics information and the way to leverage them.
> - The class statistics in IL2M is the prediction score of the classifier, while ours is the class distribution statistics in deep feature space.
> - IL2M uses the class statistics to calibrate the prediction of a continual learner in a post-processing manner, while our method leverage the statistics to automatically learn a balanced classifier at training time.
>
> We will cite this paper and discuss their relations in our final paper.
>
> **Q5: I could not understand the meaning of the numbers contained in Table 1. Are they average accuracy values or incremental accuracy values? Are these numbers referring to different experiment or do they refer to the same experiment? What dataset is being used?**
>
> Sorry for the confusion. The used dataset in Table 1 is CIFAR-100. The numbers (the number of classes learned so far) refer to the same experiments at different incremental step, i.e, 50 + 5*10 setting: the first task includes 50 classes, and each of the other 5 tasks has 10 classes. The values are the incremental accuracy. We will improve our descriptions to make this clearer in final paper.
>
> **Q6: What does “eigenvalues of covariance” mean at line 182? How exactly is the intrinsic dimension of features defined at line 188?.**
>
> - Here the covariance means the covariance matrix of deep features, and the eigenvalues are got by eigenvalue decomposition of the covariance matrix, as shown in e.q.2 on page 4.
> -  As for the intrinsic dimension of features, we follow the definition in [Ansuini, et al., 2019], i.e. the minimal number of parameters needed to describe a representation. Particularly, we follow (section3.3 in [Ansuini, et al., 2019]) to estimate the dimensionality by computing the number of components that should be included to describe 90% of the variance in the features. A recent paper [Yu, et al., 2020] uses similar estimation to study representation learning. We will introduce these relevant references and make more explanations in final paper.
>
> [Ansuini, et al., 2019] Intrinsic dimension of data representations in deep neural networks. NeurIPS. 2019.
>
> [Yu, et al., 2020] Learning Diverse and Discriminative Representations via the Principle of Maximal Coding Rate Reduction. NeurIPS 2020.
>
> **Q7: Running continual learning experiments..., What code base was used for the experiments,... Are all results computed anew for all methods or taken from other works? hyper-parameter selection...?**
>
> For our method, we built our own code, in which some modules (e.g., data loader, knowledge distillation framework) were referred to some open-source codes. For LwF-MC and LwM, we implement them using our own code. For MUC and CCIL, we use its official code. For input-replay based methods (iCaRL, EEIL and LUCIR), we run the experiments based on the codes provided by [21]. We mainly use the original hyper-parameters recommended by original papers. We are going to release our codes. All the details can be found there, and the results will be reproducible.
>
> **Q8: In the ablation study, I could not clearly understand how the baseline is constructed: how is the class-mean of old classes used to regularize the classifier (line 311)? Does this mean that only part of the semanAug loss is used?**
>
> SemanAug generates pseudo-features by performing implicit augmentation using class-means of old classes. The baseline here means that we do not generate pseudo-features using semanAug, but only replay the class-mean of each old class, when training new classes. By doing so, we aim to validate the effectiveness of semanAug compared with only replaying class-mean.
>
> **Q9: How is the forgetting measure at line 324 defined? There are several instances of such measure in CL literature, so I would need a citation to understand which one the authors are referring to.**
>
> We use the definition in [Chaudhry, et al., 2018] to measure forgetting. The reference and computation details will be included in the revision.
>
> [Chaudhry, et al., 2018] Riemannian walk for incremental learning: Understanding forgetting and intransigence. ECCV 2018.
>
> **Q10: Other suggestions: adding variance values for the presented results; the equation 3 could be simplified; In line 202, keep a neutral tone instead; the comic sans typeface is unusual in scientific papers; there are several typos that need fixing.**
>
> Thanks for your helpful and constructive suggestions to improve the clarity of our manuscript. We appreciate them and will further improve our paper accordingly. Moreover, all the typos have been corrected, and we will try our best to carefully polish and check the final paper.

---

> > ### Comment · Reviewer_9tuP · 2021-08-25
> > **rebuttal**
> >
> > I do appreciate the effort and the clarification in the replies.
> > I do consolidate my opinion that the paper is above the acceptance bar.
> > This also depends on the theoretical impact of the contribution which is indeed limited and the presentation with several typos and few inconsistencies that I hope will be fixed if the paper is accepted.

---

> > > ### Author Response · Authors · 2021-08-31
> > > **Re: rebuttal**
> > >
> > > Many thanks for replying. We will carefully check the paper to fix typos and inconsistencies.

---

### Official Review · Reviewer_2LZR · 2021-07-15

**Rating:** 4
**Confidence:** 4

**Summary:**

The paper addresses two problems which occur in the class-incremental learning: representation bias and classifier bias. To address the representation bias, the authors propose to learn a less compact representation in each task. More concretely, they augment the number of existing classes by introducing random interpolations among real samples. Regarding classifier bias, the authors propose to generate an infinite number of features from of classes to maintain the decision boundary of previously learned classes. The insights into representation learning by analyzing the structural characteristics of the learned embedding space via spectral decomposition is interesting. The paper is relatively well written, although it lacks details in some aspects. The approach is moderately novel, being a combination of existing techniques. The experimental validation does not confirm that the proposed approach improve the current state of the art in class-incremental learning.

**Ethical Concerns:**

Not applicable.

**Limitations And Societal Impact:**

The current work does not present a negative societal impact.

**Main Review:**

Here are my concerns:
1. The problems addressed in the current paper, namely representation bias and classifier bias, are not depicted clearly enough in Figure 1. I would recommend to improve the clarity of the figure.
2. Page 5, 'Feature Compression Perspective': what is the difference between the 'overall features' and 'learned features'? Why there is a difference in dimensionality in both cases?
3. Figure 4: According to section 3.1.2, the extracted features from f_(t-1) and f_t are further compressed, right? This step is not depicted in the figure.
4. An Algorithm to summarize all the steps of the approach would increase the clarity of the paper.
5. There are several typos in the paper, but most of them in page 7.


**Time Spent Reviewing:**

4 hours reading + 1 hour filling the questionnaire

---

> ### Author Response · Authors · 2021-08-10
> **Response to Reviewer 2LZR**
>
> We sincerely appreciate your thoughtful comments, efforts, and time. We respond to each of your questions and concerns one-by-one in what follows:
>
> **Q1: The approach is moderately novel, being a combination of existing techniques.**
>
> We respectably disagree our method to be a simple combination of existing techniques.
> - Our method is driven from our analysis of representation bias and classifier bias in Class-IL. To address these problems, we propose a dual augmentation framework: classAug and semanAug, which are complementary and show strong performance on Class-IL.
> - ClassAug is a new method, which is motivated by our analysis of the negative effect of regularization techniques based on spectral decomposition in section 3.1. We have discussed the difference between classAug and mixup [19] in L225-L232, and the novelty of classAug has also been recognized by other reviewers.
> - SemanAug is inspired from ISDA [24] and MCF [23], however, the motivation and implementation are quite different since we focus on Class-IL. We have discussed the difference between samanAug and ISDA in L255-L261. The effectiveness of semanAug for Class-IL demonstrates that pseudo-feature replay is a promising direction deserving of further exploration.
>
> **Q2: The experimental validation does not confirm that the proposed approach improve the current state of the art in class-incremental learning.**
>
> The comparison should be made by differentiating two types of methods: non-exemplar based and exemplar-based. Current SOTA Class-IL methods are all exemplar-based, which often store a portion of old samples (e.g., 20 samples per class in iCaRL). As shown in experimental section, without storing any old data or using complex generative models (e.g., GAN), our method can outperform non-exemplar based methods (MAS, LwF-MC, MUC, LwM) by large margins and achieve comparable results with strong exemplar-based methods. The strong performance of our approach has also been acknowledged by other reviewers. We hope to draw the attention of researchers back to the non-exemplar based method (which is more realistic in practical applications) by rethinking the necessity of storing old samples in Class-IL.
>
> **Q3: The problems addressed in the current paper, namely representation bias and classifier bias, are not depicted clearly enough in Figure 1. I would recommend to improve the clarity of the figure.**
>
> Representation bias and classifier bias are important concepts in our approach for class-incremental learning. Following your suggestion, we will further improve Fig. 1 to make it clearer, and we will also modify our descriptions to better explain these two problems.
>
> **Q4: Page 5, 'Feature Compression Perspective': what is the difference between the 'overall features' and 'learned features'? Why there is a difference in dimensionality in both cases?**
>
> The 'overall features' means the features of all training samples, while the ‘learned features for each class’ focus on the features of a specific class. Generally, the space spanned by the 'features for each class' is a subspace of the space spanned by the 'overall features'. We follow [Yu, et al., 2020] for the statement of 'overall features' and 'learned features for each class'. We will improve its clarity in final paper.
>
> [Yu, et al., 2020] Learning Diverse and Discriminative Representations via the Principle of Maximal Coding Rate Reduction. NeurIPS 2020.
>
> **Q5: Figure 4: According to section 3.1.2, the extracted features from $f_{t-1}$ and $f_{t}$ are further compressed, right? This step is not depicted in the figure.**
>
> We do not compress the extracted features from $f_{t-1}$ and $f_{t}$. As shown in Fig. 4 and Section 3.3, the extracted features from $f_{t-1}$ and $f_{t}$ are used to compute the knowledge distillation loss in L270, which is a widely used regularization to reduce forgetting in incremental learning. Section 3.1.2 focuses on the representation learning and provides a feature compression perspective to understand regularization techniques (e.g., Mixup, Cutmix, LS).
>
> **Q6: An Algorithm to summarize all the steps of the approach would increase the clarity of the paper. There are several typos in the paper, but most of them in page 7.**
>
> Thanks for your suggestion. We will add an algorithm in final paper to summarize all the steps involved in our method. We have carefully checked and corrected the typos in our paper.

---

> > ### Author Response · Authors · 2021-08-30
> > **To Reviewer 2LZR**
> >
> > Dear Reviewer 2LZR,
> >
> > We have tried our best to address all your concerns and provided clarifications on all confusing concepts. Could you please kindly re-evaluate our paper based on the current situation? If you have any further questions, we are also very glad to discuss them.
> >
> > Sincerely yours,
> >
> > Authors

---

> > > ### Comment · Reviewer_2LZR · 2021-09-02
> > > **Final decision**
> > >
> > > Despite the fact the authors addessed my questions, I can not recommend this paper for acceptance. In my opinion, the paper lacks a significant scientific contribution and the experimental results are not very convincing regarding the performance of the algorithm. Therefore, I have decided to maintain my initial rating which is 4.

---

> > > > ### Author Response · Authors · 2021-09-02
> > > > **Re: Final decision**
> > > >
> > > > **Scientific contribution.** We propose a novel dual augmentation framework, in which the classAug aims to reduce the representation bias and the semanAug focuses on the classifier bias in a complementary manner. Particularly, the two augmentations in our methods are not existing techniques, and we have discussed the novelty of them in detail in the answers to Q1. Moreover, it remains unknown what kind of representations can facilitate incremental learning. We study and answer this question via spectral decomposition for the first time.
> > > >
> > > > **Performance of the algorithm.** The performance of our method is quite strong. Please note that current SOTA Class-IL methods are all exemplar-based (e.g., requiring 20 samples per class in iCaRL and UCIR). While our method can achieve comparable results without storing any old samples. Moreover, compared with non-exemplar based methods, our approach is clearly the state-of-the-art. The strong performance of our approach has also been acknowledged by other reviewers.

---

### Official Review · Reviewer_qrF1 · 2021-07-16

**Rating:** 6
**Confidence:** 5

**Summary:**

This paper proposes a class-incremental learning (CIL) method that addresses the representation-bias and classifier-bias observed in CIL. The goal is to learn transferable ans compact representations for incremental learning and leverage the distribution of old classes to avoid forgetting and to jointly learn a unified classifier. This paper also presents an interesting finding that regularization in CIL have a negative effect.

**Limitations And Societal Impact:**

It is not clear how this method is applicable to more challenging CIL problems where there is distribution shift and out-of-distribution data. It would be useful to add this to the discussion.

**Main Review:**

This paper explores important questions in CIL: i) which parts of the representations are tend to be forgotten, ii) how regularization impacts representation learning and iii) how to facilitate representation learning in CIL. To analyze the forgetting, they use spectral decompositions, obtain the eigenvalues and eigenvectors, and check the cosine similarity between eigenvectors. To learn the transferable representations, they propose to use class augmentations. To learn the classifier, semantic augmentation is used. Class augmentation and semantic augmentation are called as dual augmentation framework and it is the proposal of this paper. To the best of my understanding, this approach is novel and has not been used before. Besides this approach doesn't require examplers and replay.

It would be useful to clarify a few points. In section 3.1.1 it is mentioned that intuitively the corresponding angle could capture the representation shift between the old and updated feature extractors during incremental learning, and reflect the forgetting along different directions in feature space. It is not clear why intuitively this is correct. The complexity of dual augmentation framework is not analyzed in the paper. The experiments show that the results of no augmentation, only semantic augmentation, only class augmentation and their combination and it seems the combination of both augmentation outperforms the others by margin, but it is not clear how costly to run the combination, which augmentation is cheaper etc.

The experiments are performed on CIFAR-100 and Tiny-ImageNet. These datasets are called challenging but it is not explained in which sense they are challenging. To the best of my knowledge, they are standard benchmarking datasets for incremental learning and it would be useful to run additional experiments on real challenging datasets where there are data samples from different domains.

The structure of the paper is nice, the problem, motivation and the goal is clear. But there are so many typos and grammatical mistakes that makes to follow the paper difficult. Some examples: p5, l161 two groups "of" eigenvectors, l170 firstly trained -> first trained, l185 simimlar -> similar, p7, l263 discribes -> describes, l266 foucs -> focus, diffrnt -> different, ... etc.

**Time Spent Reviewing:**

12

---

> ### Author Response · Authors · 2021-08-10
> **Response to Reviewer qrF1**
>
> We sincerely appreciate your thoughtful comments, efforts, and time. We respond to each of your questions and concerns one-by-one in what follows:
>
> **Q1: In section 3.1.1 it is mentioned that intuitively the corresponding angle could capture the representation shift between the old and updated feature extractors during incremental learning, and reflect the forgetting along different directions in feature space. It is not clear why intuitively this is correct.**
>
> For incremental learning, the meaning of “preserve old knowledge” refers to maintain the previously learned decision boundary among classes. At representation level, for an old class, the shape (i.e., covariance) of the deep features should not be changed too much. Therefore, we use corresponding angle, which is defined based on spectral decomposition, to measure the similarity of the shape of the feature space before and after learning new tasks. If an eigenvector direction only changes slightly after updating the feature extractor, the corresponding angle is small, and vice versa. Therefore, the corresponding angle is reasonable to reflect the forgetting along certain directions in the deep feature space.
>
> **Q2: The complexity of dual augmentation framework is not analyzed in the paper. It is not clear how costly to run the combination, which augmentation is cheaper.**
>
> ClassAug and SemanAug are complementary in our Class-IL framework, in which the former focuses on representation learning and the latter focuses on the classifier learning. As for the computational complexity, ClassAug involves input level sample mixing and the augmented samples are fed to feature extractor. Differently, SemanAug performs implicit old feature generation in deep feature space (the penultimate layer). Therefore, SemanAug is cheaper compared with ClassAug from the computation perspective. We will add the speed comparison of different methods (no augmentation, only semantic augmentation, only class augmentation and their combination) in the final paper.
>
> **Q3: The experiments are performed on CIFAR-100 and Tiny-ImageNet. These datasets are called challenging but it is not explained in which sense they are challenging.**
>
> We agree with you that CIFAR-100 and Tiny-ImageNet are standard benchmark datasets for incremental learning, and some methods have achieved remarkable performance on them under the Task-IL or input-replay based Class-IL settings.
> - However, these datasets are still quite challenging for non-exemplar based Class-IL methods which do not save old training samples (this is more realistic in practical applications), especially for long-step Class-IL (as shown in Fig.5).
> - Moreover, although some input-generation (e.g., GAN) based methods can perform well on MNIST and SVHN, they can easily fail on natural image based datasets like CIFAR-100 and Tiny-ImageNet.
>
> Our method can perform significantly better compared with other non-exemplar based methods and achieve comparable results with representative input-replay based methods on these datasets.
>
> **Q4: It is not clear how this method is applicable to more challenging CIL problems where there is distribution shift and out-of-distribution data. It would be useful to add this to the discussion.**
>
> Thanks for your suggestion. Class-IL is quite challenging if not storing old samples. When distribution shift and out-of-distribution (OOD) data are added, the difficulty of the problem will be further increased. We hypothesize that an effective Class-IL method with good OOD robustness could perform well when there are distribution shift and out-of-distribution data. Since the Class-IL ability of our method has been verified in the paper, here we conduct standard OOD detection experiments following [Hendrycks, et al., 2017] and [Lee, et al., 2018] to show that the proposed classAug can also enhance the OOD robustness of the original model, as shown in the following table (ResNet-18 on CIFAR-10 (In-distribution); OOD datasets: MNIST, Fashion-MNIST, LSUN, Tiny-ImageNet).
>
> |Metric|Method|MNIST|Fashion|LSUN|Tiny-ImageNet|
> |---|---|---|---|---|---|
> |**AUROC**|**baseline**|87.02|90.28|88.50|88.49|
> |-|**classAug**|**94.99**|**94.40**|**93.90**|**93.92**|
> |**AUPR-In**|**baseline**|79.89|86.18|83.48|83.84|
> |-|**classAug**|**93.05**|**92.43**|**91.08**|**91.77**|
> |**AUPR-Out**|**baseline**|92.26|94.26|92.92|92.70|
> |-|**classAug**|**97.20**|**96.78**|**96.73**|**96.55**|
>
> In conclusion, we agree that Class-IL with OOD data would make the method to be more applicable and useful in real situations, and this could be an interesting future direction. We will add such discussions in the paper.
>
> [Hendrycks, et al., 2017] A baseline for detecting misclassified and out-of-distribution examples in neural networks, ICLR 2017.
>
> [Lee, et al., 2018] A simple unified framework for detecting out-of-distribution samples and adversarial attacks, NeurIPS 2018.
>
>
> **Q5: The structure of the paper is nice, the problem, motivation and the goal is clear. But there are so many typos and grammatical mistakes.**
>
> Thanks for your careful reading, the typos have been corrected. We will try our best to carefully polish and check the final paper.

---

> > ### Comment · Reviewer_qrF1 · 2021-08-29
> > **Re: Q1**
> >
> > Many thanks for clarification. It would be useful for readers if you can add this explanation to the paper (at least briefly).

---

> > > ### Author Response · Authors · 2021-08-31
> > > **Re: Re: Q1**
> > >
> > > Thanks for replying. According to your suggestion, we will add the explanations in final paper.

---

### Official Review · Reviewer_cmzc · 2021-07-18

**Rating:** 6
**Confidence:** 4

**Summary:**

This work tackles incremental image classification problem, and proposes two techniques, ClassAug and SemanticAug, to improve the overall accuracy over all classes, including old and new classes. Results on two benchmarks, including CIFAR-100 and Tiny-ImageNet, are presented to demonstrate the superior performance.

**Ethical Concerns:**

I could not find ethical concerns for the proposed method.

**Ethics Review Area:**

["I don’t know"]

**Limitations And Societal Impact:**

The chance of having negative social impact is low. So it is ok authors do not address them.

**Main Review:**

Strength
  -	The method does not require to store example of old classes, and its performance is better than non-exemplar based methods and close to exemplar-based methods on two benchmarks.
  -	ClassAug seems to be an original idea with supporting evidence from spectral analysis on primary eigenvector components in the per-class features

Weakness
  -	One issue I find with the paper writing is this paper is not self-contained. In Table 1, a minimal introduction to iCaRL and CCIL is not missing, which makes it difficult to get the idea in section 3.1 before reviewing prior papers. Similarly, L169, minimal introduction to LwF-MC is not included.
  -	The proposed SemanticAug seems to be an incremental contribution as it resembles ISDA [24] a lot. While the discussions at L255 are valid, the adaptation of ISDA idea in incremental classification seems to be straight-forward.
  -	Some technical exposition is incomplete.
    o	Fig 2 (b), when data augmentation is used, such as mixup, cutmix, it often requires more training epochs to learn a better model. Can you clarify whether results with data augmentation in Fig 2 (b) are obtained with more training epochs?
    o	L217, how many augmented classes (i.e. m) are added? Any data sampling technique is used to balance original new classes and augmented new classes? how many images are generated for each augmented class?
    o	L239, for each old class, do you fix the M deep features once they are generated from a normal distribution with fixed mean/covariance? Since backbone is also updated as more new tasks are added, the mean/covariance will be outdated.
    o	L292, what is “herd” selection technique?


**Time Spent Reviewing:**

4 hours

---

> ### Author Response · Authors · 2021-08-10
> **Response to Reviewer cmzc**
>
> We sincerely appreciate your thoughtful comments, efforts, and time. We respond to each of your questions and concerns one-by-one in what follows:
>
> **Q1: In Table 1, a minimal introduction to iCaRL and CCIL is not missing, which makes it difficult to get the idea in section 3.1 before reviewing prior papers. Similarly, L169, minimal introduction to LwF-MC is not included.**
>
> Thanks. We will add more introductions for these methods in the final paper to make it self-contained. These methods (iCaRL, CCIL and LwF-MC) are representative baselines that we used to conduct the proof-of-concept experiments in section 3.1. Specifically, iCaRL [13] is a popular and representative Class-IL method that uses knowledge distillation to reduce forgetting, while CCIL [45] is a recent approach that uses sampling balance strategy to alleviate forgetting. LwF-MC [13] is a popular non-exemplar based Class-IL method based on knowledge distillation.
>
> **Q2: The proposed SemanAug seems to be an incremental contribution as it resembles ISDA [24] a lot. While the discussions at L255 are valid, the adaptation of ISDA idea in incremental classification seems to be straight-forward.**
>
> We restate our innovations and contributions as follows:
> - As we discussed in L255-L261, both our motivation and the way to leverage semanAug are different from original ISDA: to alleviate forgetting problem in Class-IL, semanAug augments old class-mean to maintain the learned decision boundary; while ISDA applies semanAug on all the training samples to improve generalization in standard supervised learning.
> - Recent Class-IL literature often use input replay or complex generative model (e.g., GAN) based data generation to address the forgetting problem. Differently, we explore pseudo-feature replay in this paper. Once we decide to use feature replay, a straight-forward way is explicitly generating pseudo-feature, as described in L239-L242. However, as described in L246-L48, we finally adopt SemanAug to achieve feature pseudo-rehearsal for high efficiency.
> - Our method consists of dual augmentations, in which SemanAug is complementary with classAug, and their combination has shown to be highly effective for Class-IL. Moreover, they are not proposed separately, but from our joint analysis of representation bias and classifier bias in Class-IL.
>
> **Q3: Can you clarify whether results with data augmentation in Fig 2 (b) are obtained with more training epochs?**
>
> Yes, the results with data augmentation in Fig 2 (b) are obtained with more training epochs as suggested by [20]. Nevertheless, we find that training all models with the same epochs obtains similar results. We will make this clear in the final paper.
>
> **Q4: L217, how many augmented classes (i.e. m) are added? Any data sampling technique is used to balance original new classes and augmented new classes? how many images are generated for each augmented class?**
>
> The number of augmented classes (i.e. m) depends on the number of (original) classes at current incremental step. Taking CIFAR-100 as an example, the m is 45 for 5 phases setting where each incremental step has 10 classes; and m is 10 for 10 phases setting where each incremental step has 5 classes. We generally use random sampling to keep the same quantity of samples for each class including original and novel classes, for the purpose of balanced training. Considering we will release source codes, those details were not included in the submission. Following your suggestion, we will add them to the manuscript.
>
> **Q5: L239, for each old class, do you fix the M deep features once they are generated from a normal distribution with fixed mean/covariance? Since backbone is also updated as more new tasks are added, the mean/covariance will be outdated.**
>
> We avoid saving the generated deep features because of the memory issue. The features are implicitly augmented during each training step. Although the backbone is updated, the fixed mean/covariance helps to restrain the original class distribution from dramatic distortion. Indeed, we have tried to transform the class-mean by learning a linear transformation or MLP. But the performance is worse, since it is hard to learn an accurate transformation without old data.
>
> **Q6: L292, what is “herd” selection technique?**
>
> “Herd” is a sample selection technique that was firstly proposed in iCaRL [13], and has become a widely used technique in input-replay based Class-IL methods [13, 21]. It selects and memorizes those samples whose deep features are close to the class mean. Intuitively, the selected samples using this technique are representative samples in each class. We will add more explanations in the paper.

---

### Author Response · Authors · 2021-08-25
**To Reviewers**

Dear reviewers,

Thanks a lot for your time and efforts in reviewing our paper. We have tried our best to address all mentioned concerns. We would appreciate it if you could take a look at our response.  As the discussion deadline is approaching, your feedback is very important to us, and if there are any new questions, we can therefore reply in time.

Sincerely yours,

Authors

---

### Decision · Program_Chairs · 2021-09-27

**Decision:**

Accept (Poster)

**Comment:**

We thank the authors for the additional clarifications provided in their rebuttal, which resolved most of the concerns raised by the reviewers. The contributions are relatively simple, but convincingly shown to be very effective compared to more sophisticated approaches recently proposed in continual learning. These results provide new insights and will benefit the NeurIPS community. However, the reviewers also provided a list of improvements that should be incorporated in the revised version. The paper was not self-contained at times and needs to be polished in order not to hamper the understanding. Exposition of the experiments would also benefit of a revision.